# Finite-temperature symmetric tensor network for spin-1/2 Heisenberg antiferromagnets on the square lattice

**Didier Poilblanc[1][⋆], Matthieu Mambrini[1] and Fabien Alet[1]**

**1** Laboratoire de Physique Théorique, C.N.R.S.
and Université de Toulouse, 31062 Toulouse, France

⋆ didier.poilblanc@gmail.com

## Abstract

Within the tensor network framework, the (positive) thermal density operator can be approximated by a double layer of infinite Projected Entangled Pair Operators (iPEPO) coupled via ancilla degrees of freedom. To investigate the thermal properties of the spin-1/2 Heisenberg model on the square lattice, we introduce a family of fully spin-$SU(2)$ and lattice-$C_{4v}$ symmetric on-site tensors (of bond dimensions $D = 4$ or $D = 7$) and a plaquette-based Trotter-Suzuki decomposition of the imaginary-time evolution operator. A variational optimization is performed on the plaquettes, using a full (for $D = 4$) or simple (for $D = 7$) environment obtained from the single-site Corner Transfer Matrix Renormalization Group fixed point. The method is benchmarked by a comparison to quantum Monte Carlo in the thermodynamic limit. Although the iPEPO spin correlation length starts to deviate from the exact exponential growth for inverse-temperature $\beta \gtrsim 2$, the behavior of various observables turns out to be quite accurate once plotted w.r.t the inverse correlation length. We also find that a direct $T = 0$ variational energy optimization provides results in full agreement with the $\beta \to \infty$ limit of finite-temperature data, hence validating the imaginary-time evolution procedure. Extension of the method to frustrated models is described and preliminary results are shown.



# 1  Introduction

The spin-1/2 Heisenberg antiferromagnet on the square lattice orders magnetically at zero temperature but, compared to the classical Néel state, quantum fluctuations induce a reduction of the order parameter [1]. The value of the staggered magnetization is about 60% of its classical counterpart (1/2) [2–5]. At any non-zero temperature, Mermin-Wagner theorem implies the restoration of the full continuous spin-SU(2) symmetry but the magnetic correlation length diverges exponentially fast when approaching zero temperature [6]. These features have been well established by quantum Monte Carlo (QMC) simulations [2,4,7,8] which are free of any sign problem in the absence of magnetic frustration.

In the last decade, tensor network methods have brought considerable new insights in the understanding of models of interacting quantum spins. At zero temperature, two-dimensional Projected Entangled Pair States (PEPS) have provided many valuable ansatze for groundstates satisfying the area-law of the entanglement entropy [9–13]. Although small deviations from the area-law are expected [14,15], simple PEPS with small bond dimensions offer good approximate realizations of the symmetry-broken Néel state [16,17]. However, describing SU(2)-symmetric groundstates with critical or even very long-range correlations remains an open challenge [18].

The implementation of finite-temperature tensor network algorithms in two-dimensions has been steadily developed and improved over the last ten years [19–22], and benchmarked on simple models as the quantum Ising model [20,21,23,24], the quantum compass model [25] or the Shastry-Sutherland Heisenberg model [26, 27]. These methods are based on a parametrization of the thermal density operator (TDO) in terms of a Projected Entangled Pair Operator (PEPO). Among the recent developments, the most efficient framework uses a double-layer PEPO, instead of a single layer one, which naturally guarantees the positivity of the approximated TDO. The PEPO formalism also allows to construct the TDO via an imaginary-time evolution [28–31]. The method can be used directly in the thermodynamic limit with infinite-PEPO (iPEPO). Its efficiency is remarkably good, provided the phase is gapped. Here, our aim is to test this method in the more complex case of a gapless phase, with a diverging correlation length at low temperature. Interesting new features of this work, which have not been explored before at finite temperature, are the implementation of $SU(2)$ spin and $C_{4v}$ lattice symmetric tensors, and plaquette-type update approaches.

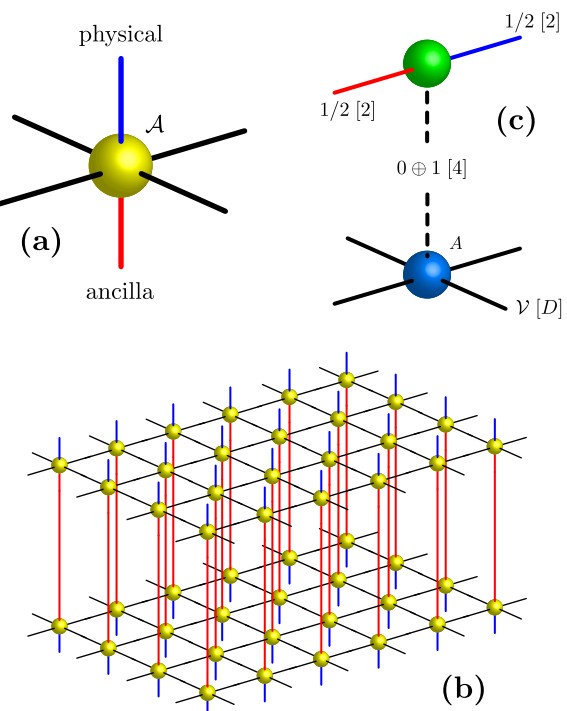

Figure 1: (a) Elementary SU(2)-symmetric $C_{4v}$-symmetric on-site tensor $\mathcal{A}$. (b) Thermal density operator written as an (infinite) double layer. A thermal expectation value is obtained by applying local operators on the physical (blue) legs and tracing over all degrees of freedom. (c) Construction of the rank-6 local tensor by contracting rank-5 and rank-3 SU(2)-symmetric tensors. The dotted line stands for the 4-dimensional internal space given by the direct sum of the singlet (spin 0) and triplet (spin 1) SU(2) multiplets.

## 2 Symmetric iPEPO method

### 2.1 Thermal density operator framework

The method is based on a simple tensor network approximation of the (unnormalized) thermal density operator $\rho(\beta) = \exp(-\beta H)$ at temperature $T = 1/\beta$, as depicted in Figs. 1. The building block of this construction is the rank-6 on-site tensor $\mathcal{A}$ of Fig. 1(a) containing 4 virtual bonds of dimension $D$ to be connected to the neighboring sites, a physical bond (in blue) carrying the $d = 2$ spin-1/2 degrees of freedom and an ancilla bond (in red) of same dimension. The square lattice translation and point group symmetry can be enforced by assuming identical $C_{4v}$-symmetric tensors on every site. In addition, since spin SU(2)-symmetry is preserved at any *finite* temperature, we also enforce the SU(2)-symmetry at the level of the on-site tensor (see later for details). An infinite Projected Entangled Pair Operator (iPEPO) $\mathcal{L}$, i.e. a linear map from the (infinite) physical space to the (infinite) ancilla space, is obtained by contracting the tensor network over all virtual degrees of freedom[1]. The thermal density operator can then be approximated by a double layer of such iPEPOs, $\mathcal{L}\mathcal{L}^T$, after contracting over all ancillas as shown in Fig. 1(b). It has been pointed out that such a construction naturally guarantees the positivity of the (approximate) TDO. Note that this approximative representation of the TDO becomes more and more accurate for increasingly larger (virtual)

---

[1]To avoid the insertion of matrices on the lattice bonds, a 180° spin rotation is performed on one of the two sublattices.

Table 1: Numbers of elementary tensors in each of the tensor classes defined by their virtual spaces $\mathcal{V}$. The number of elementary $C_{4v}$-symmetric tensors are shown in the third column. When the site $C_{4v}$ point group symmetry is broken to $C_s$, the $N_D$ tensors are "split" into $N_D'$ tensors (fourth column). For $D = 7$, assuming "color" spin-1 exchange symmetry reduces the number of elementary tensors, as shown by the numbers in parenthesis.

| $\mathcal{V}$ | $D$ | $N_D$ | $N_D'$ |
|---|---|---|---|
| $0$ | $D = 1$ | 1 | 1 |
| $0 \oplus 1$ | $D = 4$ | 8 | 13 |
| $0 \oplus \frac{1}{2} \oplus 1$ | $D = 6$ | 21 | 45 |
| $0 \oplus 1 \oplus 1$ | $D = 7$ | 49 (31) | 115 (73) |

bond dimension $D$. With the iPEPO approximation of the TDO at hand, any thermal average can, in principle, be computed by applying a local operator (or a string of operators) on the (e.g. top) physical layer and tracing over all top and bottom physical degrees of freedom.

*Symmetric PEPO ansatz* - Here, by enforcing the lattice and spin symmetries of the problem, we simplify the ansatz further since the on-site tensor can then be written as a linear superposition of a small number $N_D$ of (fixed) symmetric tensors $\mathcal{T}_\alpha$,

$$\mathcal{A}(\beta) = \sum_{\alpha=1}^{N_D} c_\alpha(\beta) \mathcal{T}_\alpha, \tag{1}$$

where $c_\alpha(\beta)$ are real *temperature-dependent* coefficients. The temperature dependence of the TDO $\rho_\beta$ is therefore only encoded in the temperature dependence of the latter coefficients.

The construction of symmetric PEPO is a generalization of the standard method introduced to derive symmetric Projected Entangled Pair State (PEPS) tensors [32]. In fact, as depicted in Fig. 1(c), the family of tensors $\mathcal{T}_\alpha$ can be obtained by contracting two families of PEPS tensors over their "physical" space $0 \oplus 1$. The family of rank-5 $A$ tensors is constructed assuming spin-SU(2) and $C_{4v}$ symmetry w.r.t. exchange of the four virtual bonds (thick segments). There are two rank-3 tensors corresponding to each fusion outcomes – spin-0 or spin-1 – of the physical and ancilla degrees of freedom, and therefore two independent subclasses of $A$ tensors obtained by contracting each of the two rank-3 tensors with the appropriate (matching) rank-5 tensors.

The families of $\mathcal{A}$ tensors are defined by the vector space $\mathcal{V}$ of the virtual degrees of freedom written as a direct sum of SU(2) irreducible representations (irreps). At infinite temperature, i.e. $\beta = 0$, the unnormalized TDO reduces to the identity and the virtual space reduces to a single singlet state, $\mathcal{V}_0 = 0$, $D = 1$. At non-zero $\beta$ we increase the entanglement between sites by introducing more SU(2) spins into $\mathcal{V}$, and the physical Hilbert space becomes the direct sum of SU(2) multiplets such as $\mathcal{V} = 0 \oplus 1$ ($D = 3$), $\mathcal{V} = 0 \oplus \frac{1}{2} \oplus 1$ ($D = 6$) or $\mathcal{V} = 0 \oplus 1 \oplus 1$ ($D = 7$). The number of $C_{4v}$ SU(2) symmetric tensors $\mathcal{T}_\alpha$ in each class is shown in the third column of Table 1.

## 2.2 Plaquette Trotter-Suzuki decomposition

With the framework established, one needs a reliable method to obtain the $\beta$-dependence of the coefficients $c_\alpha(\beta)$. We shall use here a Trotter-Suzuki decomposition. The usual initial starting point consists in splitting the unnormalized TDO into a large number $N_\tau$ of imaginary-time evolutionary steps, $\rho(\beta) = [\exp(-\tau H)]^{N_\tau}$, where $\tau = \beta/N_\tau$ is a small imaginary-time step, and start from the infinite-temperature i.e. $\beta = 0$ limit.

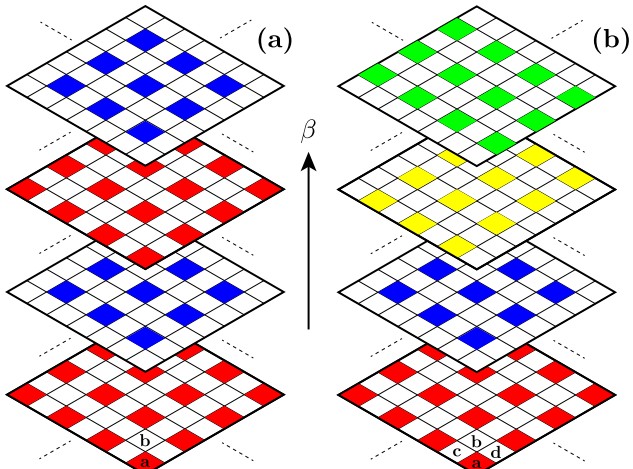

Figure 2: *a-b* (a) and *a-b-c-d* (b) checkerboard Trotter-Suzuki (TS) decomposition. Each plane corresponds to a given (imaginary) time step.

In our framework, assuming that, at a given temperature $T = 1/\beta$, $\rho_\beta$ is in the (approximate) iPEPO form, the problem reduces to evaluate the action of the "infinitesimal" propagator $G(\tau) = \exp(-\tau H)$, i.e. $\rho(\beta + \tau) = G(\tau)\rho(\beta)$, and to approximate the result as a new iPEPO defined by an updated tensor $\mathcal{A}(\beta + \tau)$ corresponding to the set of coefficients $\{c_\alpha(\beta+\tau)\}$. To realize this, in practice, one can rewrite the Hamiltonian as $H = \sum_a \mathcal{H}_a + \sum_b \mathcal{H}_b$ or $H = \sum_a \mathcal{H}'_a + \sum_b \mathcal{H}'_b + \sum_c \mathcal{H}'_c + \sum_d \mathcal{H}'_d$, where $\{a\}$, $\{b\}$, $\{c\}$ and $\{d\}$, are the sets of plaquettes depicted in Fig. 2. Focussing on the *a-b* decomposition for the moment (which is legitimate if the Hamiltonian contains only nearest-neighbor interactions), we get $G(\tau) \simeq \prod_b G_b(\tau) \prod_a G_a(\tau)$, up to small $\mathcal{O}(\tau^2)$ Trotter errors.

## 2.3 Variational optimization of the plaquette fidelity

### 2.3.1 First step: $a$ plaquettes

Let us first concentrate on the first product term $\prod_a G_a(\tau)$ of $G(\tau)$. Since all $G_a$ plaquette propagators commute with each other, it is sufficient to consider the action of one of them to obtain the new tensors on a single plaquette, and then update the tensors on the whole lattice. Following the notation of Fig. 3(a,b), we can rewrite $G_a(\tau) = \mathcal{G}^2$ and approximate $\rho(\beta) = \mathcal{L}\mathcal{L}^T$. Up to a cyclic permutation of the operators (which does not change thermal expectation values) we obtain $\rho(\beta + \tau) = (\mathcal{L}\mathcal{G})(\mathcal{L}\mathcal{G})^T$ which we have to approximate as $\rho(\beta + \tau) \simeq \mathcal{L}'\mathcal{L}'^T$, where $\mathcal{L}'$ is a single layer iPEPO which differs from $\mathcal{L}$ only on the four sites of a single plaquette, by a new $\mathcal{A}'$ tensor. Since the action of $\prod_a G_a(\tau)$ breaks the square lattice $C_{4v}$ symmetry into $C_s$ (only the reflections w.r.t the $a$ plaquette diagonals remain), the $\mathcal{A}'$ tensor belongs to an enlarged class,

$$\mathcal{A}'(\beta + \tau) = \sum_{\alpha=1}^{N'_D} c'_\alpha(\beta + \tau)\mathcal{T}'_\alpha, \qquad (2)$$

where the new class dimensions $N'_D$ are listed in the fourth column of Table 1.

Using the purified PEPS notation of Fig. 3(c), one can define fidelities (or "overlaps") as $\langle \mathcal{L}|\mathcal{L}'\rangle = \text{Tr}[\mathcal{L}\mathcal{L}']$ or $\langle \mathcal{L}|\mathcal{G}|\mathcal{L}'\rangle = \text{Tr}[\mathcal{L}\mathcal{G}\mathcal{L}']$, where the trace is performed on the ancilla degrees of freedom. Outside of the active $2 \times 2$ plaquette, all fidelities involve the same uniform tensor network of on-site $C_{4v}$-symmetric double-layer $\mathcal{A}\mathcal{A}^T$ tensor contracted over both physical and ancilla degrees of freedom. We have used a single-site (symmetric) Corner Transfer Matrix Renormalization Group (CTMRG) [33–35], more specifically its single-site symmetric

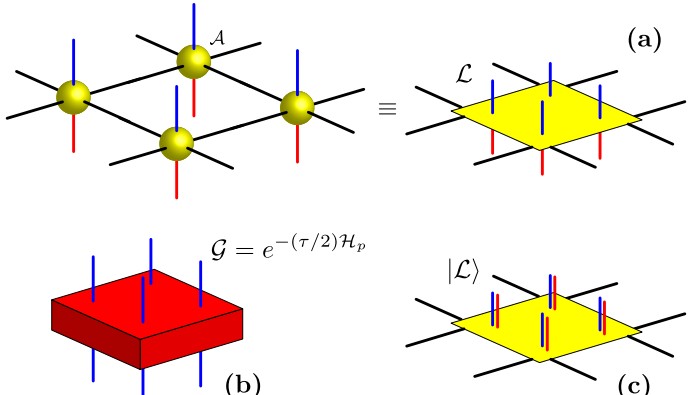

Figure 3: (a) Elementary plaquette representation of the (half-) density operator $\mathcal{L}$. (b) Plaquette gate $\mathcal{G}$. (c) The $\mathcal{L}$ iPEPO operator can be viewed as a purified PEPS $|\mathcal{L}\rangle$.

version [32], to contract the network around the active plaquette, resulting into a converged (so-called "fixed-point") SU(2)-symmetric environment of adjustable bond dimension $\chi$, as shown e.g. in Fig. 4(a).

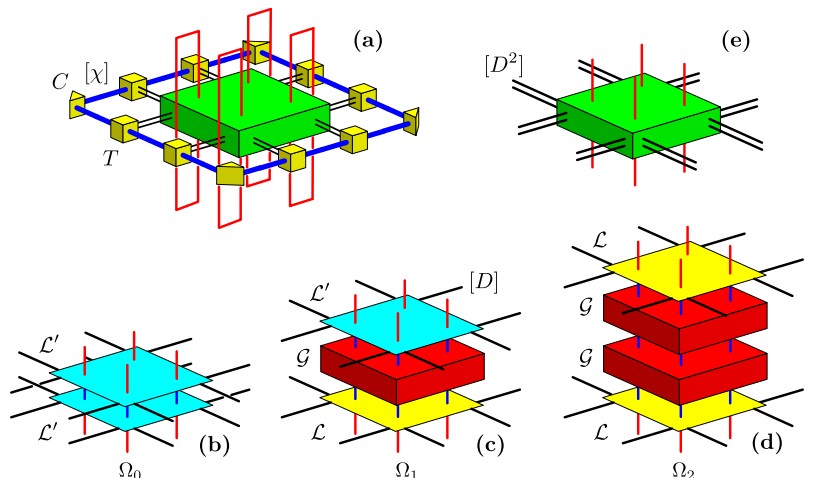

Figure 4: (a) Graphical representation of the overlaps $\omega_0$, $\omega_1$, and $\omega_2$, obtained by using the CTMRG fixed-point environment, and by contracting the operators $\Omega_0$ (b), $\Omega_1$ (c) and $\Omega_2$ (d), respectively, over the ancilla degrees of freedom and drawn in (a) using the same generic green box (e).

Minimizing the distance $\|\mathcal{G}|\mathcal{L}\rangle - |\mathcal{L}'\rangle\|$ between $\mathcal{G}|\mathcal{L}\rangle$ and $|\mathcal{L}'\rangle$ is equivalent to maximize the (normalized) fidelity

$$\mathcal{O} = \omega_1/[\omega_0 \omega_2]^{1/2}, \tag{3}$$

where the overlaps $\omega_0$, $\omega_1$ and $\omega_2$ are defined as

$$\omega_0 = \langle \mathcal{L}'|\mathcal{L}'\rangle, \tag{4}$$

$$\omega_1 = \langle \mathcal{L}|\mathcal{G}|\mathcal{L}'\rangle, \tag{5}$$

$$\omega_2 = \langle \mathcal{L}|\mathcal{G}\mathcal{G}|\mathcal{L}\rangle, \tag{6}$$

and graphically represented in Fig. 4. We have used a Conjuguate Gradient (CG) optimization

routine to maximize $\mathcal{O}$ w.r.t. the set of coefficients $\{c'_\alpha(\beta + \tau)\}$ defining the $\mathcal{A}'$ tensor located on the four sites of the $2 \times 2$ plaquette of $|\mathcal{L}'\rangle$.

### 2.3.2 2nd step: $b$ plaquettes

Once the optimum $\mathcal{A}'$ tensor is obtained on the $2 \times 2$ plaquette, it is updated on all the other lattice sites. We then have to consider the new fidelity $\mathcal{O}$ defined on a $2 \times 2$ $b$ plaquette by replacing $\mathcal{A} \rightarrow \mathcal{A}'$ and $\mathcal{A}' \rightarrow \mathcal{A}''$, in the bottom and top layers respectively, and optimize it w.r.t. the new $C_s$-symmetric tensor $\mathcal{A}''$. Applying $\mathcal{G}$ on a $b$ plaquette, instead of an $a$ plaquette, amounts in fact to rotate the four (bottom) $\mathcal{A}'$ tensors by $180^o$. We know that, in the limit $\tau \rightarrow 0$, the optimum $\mathcal{A}''$ tensor should become exactly $C_{4v}$-symmetric, and the small deviation from $C_{4v}$-symmetry is due to the non-commutativity of $G_a$ and $G_b$ on neighboring plaquettes and, hence is part of the Trotter error. We can partially correct it by symmetrizing the $\mathcal{A}''$ tensor and update it on the whole lattice for the next Trotter step.

### 2.3.3 Full environment versus simplified environment

At this stage, it is useful to specify how the overlaps $\omega_i$ are computed. As mentioned above, we use a single-site CTMRG algorithm to contract the network around the active plaquette. This procedure is parametrized by the number $\chi$ of (virtual) states kept at each step of truncation of the corner transfer matrix (see Ref. [32, 35] for details). In the course of the imaginary-time evolution (TE), we have used a fixed $\chi = \chi_{\text{TE}}$, typically set to $D^2$ or $2D^2$, to compute with CTMRG the new converged $C$ and $T$ environment tensors (see Fig. 4(a)) at each Trotter step. For $D = 4$, we have used a full environment (FE) scheme, meaning that the environment used to construct the $\omega_i$ overlaps retain all the $\chi^2$ and $\chi^2 D^2$ components of the fixed-point $C$ and $T$ tensors. In contrast, for $D = 7$, we used a simplified environment (SE) where we used only the largest weight of the fixed-point $\chi \times \chi$ corner matrix (and the corresponding $D^2$ components of the $T$ tensor) to construct the environment.

Note however that, in both cases, i) the CTMRG involves all $\chi = \chi_{\text{TE}}$ degrees of freedom on the bonds (in this sense the SE scheme is more elaborate than the simple update scheme [36] for which no CTMRG is performed) and ii) the same type of "brute force" variational optimization w.r.t. the components of the $\mathcal{A}'$ (or $\mathcal{A}''$) tensor is performed via a CG algorithm, necessitating repeated computations (a few thousand times at each Trotter step) of the overlaps $\omega_i$. Also, once the $\mathcal{A}'$ and $\mathcal{A}''$ tensors have been optimized, all observables (like the energy density) are computed using the full environment. We point out that, to compute the overlap cost-function, one can also consider an intermediate case, between the SE and the FE cases, by considering a number $\chi_{\text{overlap}}$ of highest weights of the fixed-point corner matrix between 1 (the SE case) and $\chi_{TE}$ (the FE case), to build a simplified environment of the $\omega_i$. Note that we have not attempted to consider such an intermediate scheme so far, it is left for future studies on the frustrated model (see below).

## 3 Results

### 3.1 Heisenberg model and comparison to QMC

We now focus on the spin-1/2 Heisenberg model on the square lattice,

$$H = J_1 \sum_{\langle i,j \rangle} \mathbf{S}_i \cdot \mathbf{S}_j, \tag{7}$$

involving only an antiferromagnetic coupling $J_1$ (set to 1) between nearest-neighbor sites $\langle i,j \rangle$, enabling a direct comparison to quantum Monte Carlo (QMC) results. In that case, the simplest

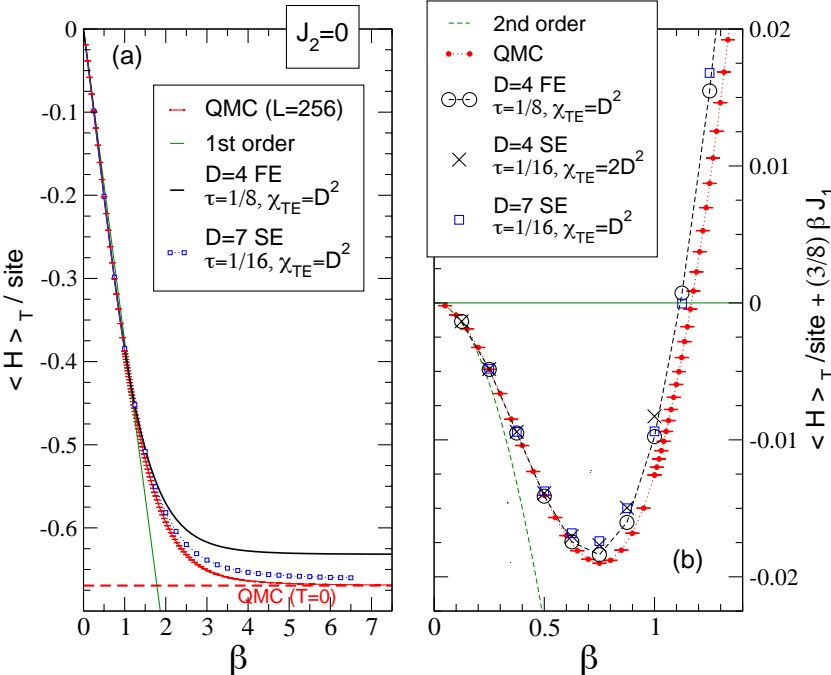

Figure 5: (a) Mean energy vs $\beta = 1/T$ computed using $\mathcal{V} = 1 \oplus 0$, $D = 4$ ($\mathcal{V} = 1 \oplus 0 \oplus 1$, $D = 7$), and FE (SE), and compared to QMC (on a cluster of size $L = 256$). The exact 1st order high-T behavior – linear in $\beta$ (green line) – is subtracted in (b) to magnify the scale and compare different imaginary-time evolution dynamics (see text) to the second order high-T correction and to QMC. The finite-temperature QMC data are to be considered in the thermodynamic limit for $\beta \leq 4$ (see text). The $T = 0$ QMC data are taken from Ref. [4].

*a-b* Trotter-Suzuki decomposition of Fig. 2(a) is possible, since all the Hamiltonian terms are contained within the *a* and *b* plaquettes. The QMC computations, which are statistically exact, are performed at non-zero temperature on finite albeit large samples of linear size *L*, thanks to the efficient loop algorithm [37]. When the sample size *L* is several times larger than the correlation length $\xi$ (one usually considers $L > 6\xi$), the QMC results can be considered to be in the thermodynamic limit, thus allowing to benchmark the iPEPO results. QMC simulations are performed on systems of size up to $L = 256$ (with $N = L^2$ spins), leading to the absence of finite-size effects down to inverse temperature $\beta = 4$ (where $\xi \simeq 40$). The correlation length $\xi$ is estimated from the second moment estimator [38, 39] using a loop improved estimator [37, 39].

### 3.1.1 Thermal energy density

As a first benchmark we computed the temperature dependence of the mean energy (per site), $e = \langle H \rangle_T / N$ where the thermal average is defined as usual as $\langle \cdots \rangle_T = \mathrm{Tr}(\rho_\beta \cdots)/\mathrm{Tr}(\rho_\beta)$, using the same $\chi = \chi_{\mathrm{TE}}$ as for the overlaps $\omega_i$. The results are shown in Fig. 5(a) for the two ansätze corresponding to $D = 4$ and $D = 7$, with small time steps for which Trotter errors are almost negligible. At high temperatures, say for $\beta < 1.5$, the iPEPO data follow very closely the QMC data[2]. Note that, already for $\beta > 0.5$ strong deviations occur w.r.t second order high-temperature expansion, $e(\beta) = -\frac{3}{8}J_1^2\beta - \frac{3}{32}J_1^3\beta^2 + o[\beta^3]$, which is picked-up relatively well

---

[2]Within QMC, for $\beta < 2$ ($\beta < 4$), finite size effects become negligible on $64^2$ ($256^2$) clusters, and results in the thermodynamic limit are obtained.

by our method. This is clear from Fig. 5(b) where the first-order, linear in $\beta$, contribution has been subtracted off to zoom in on the small energy difference: above $\beta \sim 0.5$ the iPEPO data deviate strongly from the second-order prediction while staying quite close to the exact QMC data. We believe that the small remaining deviations still observed for $D = 7$ in the range $0.5 \leq \beta \leq 1.5$ are due to the approximate SE procedure used in that case to compute the fidelity. When lowering the temperature further, the energy density starts to saturate above the exact ground-state energy $e_0^{\text{QMC}} \simeq -0.6694$ (see Ref. [4]), around $-0.636$ and $-0.661$ for $D = 4$ and $D = 7$, respectively. Note that the ansatz with $0 \oplus \frac{1}{2} \oplus 1$ ($D = 6$) is not providing a significant improvement compared to $D = 4$, so we shall not consider it further.

### 3.1.2 Spin correlation function and correlation length

At any non-zero temperature, the spin-spin correlations are short-range (from Mermin-Wagner theorem) but it is known that the correlation length $\xi$ associated to the long-distance exponential decay rises exponentially fast at (not so) large inverse-temperature $\beta$. For example, at $\beta = 5$ (respectively $\beta = 6$), $\xi$ is already beyond 120 (respectively 400) lattice spacings [6–8]. To investigate how good this feature is captured by our iPEPO approach we have computed the spin-spin correlation function $C(|i-j|) = \langle \mathbf{S}_i \cdot \mathbf{S}_j \rangle$ versus distance $r_{ij} = |i-j|$, and extracted the correlation length $\xi$ from an exponential fit $\exp(-r_{ij}/\xi)$ of the long distance behavior[3]. Although the tensor optimization is always realized at $\chi = \chi_{\text{TE}} = D^2$, converged results for the correlations require larger values of $\chi$, as we have carefully checked. Results for correlation functions obtained with $D = 4$ and $D = 7$ are reported in Figs. 6(a) and 6(b), respectively. Although, at distance $r = 1$, the spin correlation converges quickly with increasing $\beta$, strong changes with $\beta$ occur at intermediate and long distances, reflecting the rapid increase of the correlation length.

As shown in Fig. 6(c), showing the scaling of the correlation length $\xi$ with the environment dimension $\chi$, larger values of $\chi$ are needed for convergence at larger $\beta$ and at larger $D$ as well. At high temperature, the iPEPO correlation length reported in Fig. 6(d) increases linearly with $\beta$, tracking very well the QMC results. However, above $\beta \sim 2$, the QMC correlation length shoots up while the $D = 4$ iPEPO correlation length starts to saturate around 7 lattice spacings. Even for larger $D = 7$, deviations from QMC still occur quickly at intermediate $\beta$. For example, while the (exact) QMC correlation length at $\beta = 3$ ($\beta = 4$) is around 12 (39) lattice spacings, we get $\xi \simeq 4.2$ ($\xi \simeq 5.6$) and $\xi \simeq 5.5$ ($\xi \simeq 9.2$) for $D = 4$ and $D = 7$, respectively.

The iPEPO spin correlations at short and intermediate distances have been directly compared to QMC results, obtained on system sizes for which finite size effects are negligible. As shown in Fig. 6(a), a good agreement with the $D = 4$ results is obtained up to $\beta = 1.5$, while significant deviations occur at $\beta = 2$. Even for $D = 7$, strong deviations occur above $\beta \sim 2$, as seen in Fig. 6(b) comparing results at $\beta = 3$ and $\beta = 4$. Interestingly, we observe that QMC results at $\beta = 3$ agree with iPEPO results at $\beta = 4$. This suggests that one may define an effective inverse-temperature $\beta_{\text{eff}}(\beta, D) < \beta$ such that local observables computed at $\beta$ approximately match their exact values at $\beta_{\text{eff}}$. Of course, this would imply an equivalence between the two TDO (not only between specific observables) which is difficult to prove.

### 3.1.3 Finite correlation length scaling

Although the iPEPO correlation length deviates strongly from the exact behavior at intermediate and low temperatures, qualitative agreement between iPEPO and QMC still survives for some observables providing one considers their behaviors w.r.t. the inverse correlation length rather than w.r.t. the inverse-temperature. Such inverse correlation length scaling has been

---

[3]The correlation length $\xi$ can also be obtained from the spectrum of the transfer matrix, providing identical results.

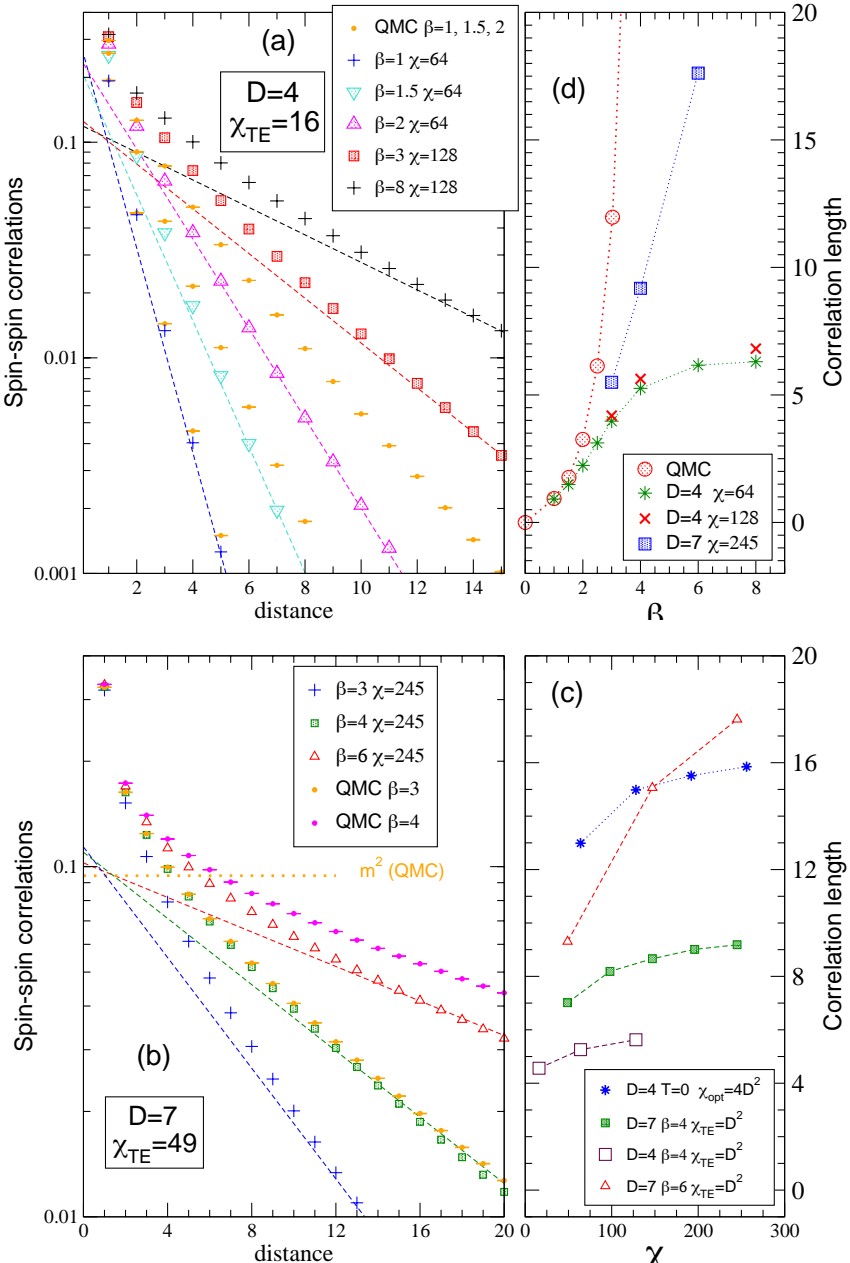

Figure 6: (a) Spin correlation function versus distance in semi-log scales, for $D = 4$ (a) and $D = 7$ (b) tensors optimized with $\chi_{TE} = D^2$. Exponential fits of the long distance behavior (not shown) are indicated as dashed lines. Converged QMC results are also shown, obtained on $64 \times 64$ ($\beta = 1, 1.5$ and 2 data displayed from bottom to top) (a) and $256 \times 256$ systems (b). $m^2$ corresponds to the $T = 0$ long-distance correlation. (c) Scaling of the spin correlation length w.r.t. $\chi$, for a few values of the temperature and $D = 4, 7$. The largest available values of $\chi$ are used in (a) and (b). Extracted spin correlation lengths are plotted vs $\beta$ in (d), and compared to QMC. Note that in (b) and (d), the data for $D = 7$ and $\beta = 6$ are not fully converged in $\chi$.

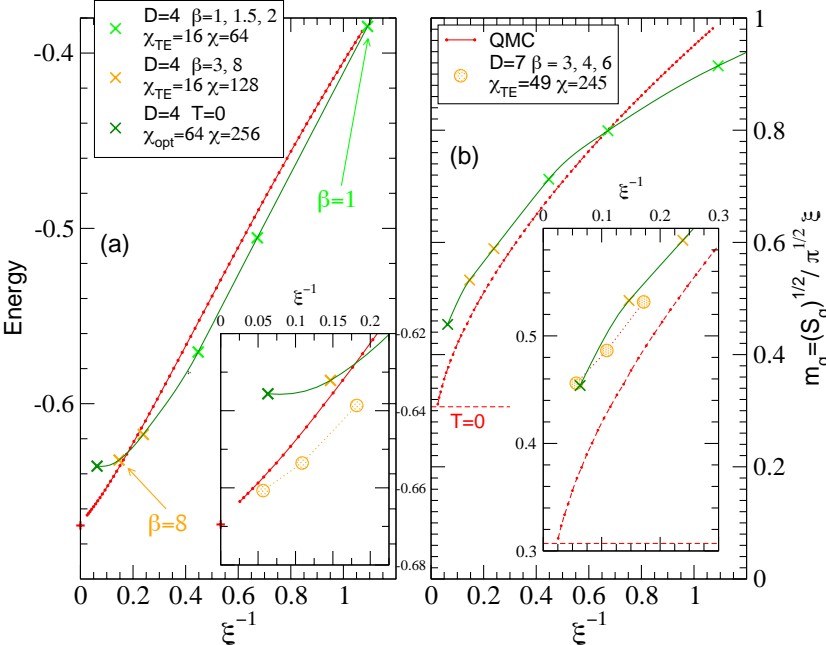

Figure 7: Energy (a) and magnetization estimator $\overline{m_q}$ (b) plotted versus the inverse of the spin correlation length, for $D = 4$ tensors optimized with $\chi_{\text{TE}} = D^2$. Large enough environment dimensions $\chi$ are chosen to get converged results (in contrast to results obtained with $\chi = \chi_{\text{TE}}$, also shown). End-points obtained directly at $T = 0$ (see text) are also included. QMC results in the thermodynamic limit are shown in the range $\beta \in [1, 4]$, for comparison, as well as the $T = 0$ magnetization [4] (horizontal dashed line in (b)). Similar results for $D = 7$ are also shown as circles in the inset of panel (a) and in panel (b).

introduced as a powerful tool for zero-temperature symmetry-broken groundstates [40, 41] and used recently to extract quantitatively the staggered magnetization curve of the frustrated quantum Heisenberg model [17]. Also, it has been applied to the case of second-order finite-temperature phase transitions associated to the spontaneous breaking of a discrete symmetry, as in the 2D quantum Ising model [42].

We investigate here whether the same concept still holds in the more challenging case of a zero-temperature phase transition associated to a continuous-symmetry breaking. Fig. 7(a) shows the behavior of the mean energy versus $\xi$-inverse where different temperature ranges have been considered for iPEPO and QMC, $\beta \in [1, 8]$ and $\beta \in [1, 4]$, respectively. The QMC energy varies almost linearly with $\xi^{-1}$ in rough agreement with the iPEPO data. Note that, although deviations do occur, such a qualitative agreement is nevertheless rather unexpected considering the very important mismatch of the correlation lengths at intermediate temperature.

We have also computed an estimator of the (zero-temperature) staggered magnetization $m_q = |\langle \frac{1}{N} \sum_i \exp(i\mathbf{q} \cdot \mathbf{r}_i) \mathbf{S}_i \rangle_0|$, with $\mathbf{q} = (\pi, \pi)$, in terms of the spin structure factor $S_q = \frac{1}{N} \sum_{ij} \exp(i\mathbf{q} \cdot \mathbf{r}_{ij}) \langle \mathbf{S}_i \cdot \mathbf{S}_j \rangle = \sum_{\mathbf{r}} = (-1)^r \langle \mathbf{S}_0 \cdot \mathbf{S}_{\mathbf{r}} \rangle$. When $\beta \to \infty$, $S_q$ should diverge as $\xi^2$ i.e. as the area of (antiferromagnetically) correlated spins. More precisely, if we crudely assume that spins are correlated, i.e. $\langle \mathbf{S}_0 \cdot \mathbf{S}_{\mathbf{r}} \rangle \simeq (-1)^r (m_q)^2$, within a disk of radius $\xi$ and that $\langle \mathbf{S}_0 \cdot \mathbf{S}_{\mathbf{r}} \rangle \simeq 0$ beyond, we then obtain

$$m_q \simeq S_q / \pi^{1/2} \xi, \tag{8}$$

when $\xi \gg 1$. The quantity $\overline{m_q} = S_q/\pi^{1/2}\xi$ on the right-hand side of (8) gives then a finite-temperature estimator of the (zero-temperature) staggered magnetization $m_q$. Note that, because of the abrupt separation assumed between correlated and uncorrelated spins, the r.h.s is approximate, even in the $\beta \to \infty$ limit, and the exact relation between the two quantities should involve a numerical prefactor (close to 1). We have plotted $\overline{m_q}$ versus $1/\xi$ in Fig. 7(b) and compared it to the QMC estimate. Although the agreement is still rough, our data extrapolated to $\xi^{-1} = 0$ clearly indicates that the zero-temperature staggered magnetization is indeed finite.

### 3.1.4 Zero-temperature limit

So far the optimization of the finite-T site tensor $\mathcal{A}$ has been performed via imaginary-time propagation. An alternative method would involve the direct variational optimization of the free energy (per site) $f = e - Ts$, where $s$ is the thermodynamic entropy per site. Computing $s$ is a notoriously hard problem for tensor networks but, at $T = 0$, it becomes feasible to minimize the thermal energy $e = \frac{1}{N}\text{Tr}(H\rho(\mathcal{A}))$ which only involves the local tensor $\mathcal{A}$ to optimize upon. We have accomplished this task using the same global optimization scheme as used for optimizing iPEPS [18, 43]. Our $T = 0$ result for $D = 4$ reveals a quite large spin correlation length, around 16 lattice spacings, as shown in Fig. 6(c). For $D = 7$, our CTMRG failed to produce a SU(2)-invariant boundary, probably because the correlation length becomes very large in the course of the optimization inducing spontaneous symmetry breaking. We have also included the $T = 0$ data point obtained for $D = 4$ to Figs. 7(a) showing that the finite-T data seem to approach, for larger and larger $\beta$, the $T = 0$ "end-point". This provides a check that our imaginary-time propagation method remains reliable at low temperature in producing the optimal TDO within its variational manifold.

Note that, in Figs. 7(a,b), we have not shown any QMC data beyond $\beta = 4$, value of $\beta$ at which finite size effects start to come into play on the largest $256 \times 256$ cluster. In the regime $\beta \gg 1$, one expects the energy to behave as $e = e_0 + C\beta^3$ [4, 6, 7] and then to approach the $T = 0$ limit $e_0$ as $-1/\ln^3(\xi^{-1})$ i.e. with a vertical slope in Fig. 7(a).

## 3.2 Extention to frustrated models

Lastly, we introduce frustrating interactions. The Hamiltonian takes the form :

$$H = J_1 \sum_{\langle i,j \rangle} \mathbf{S}_i \cdot \mathbf{S}_j + J_2 \sum_{\langle\langle k,l \rangle\rangle} \mathbf{S}_k \cdot \mathbf{S}_l, \tag{9}$$

where the antiferromagnetic $J_2$ interaction couples all next-nearest-neighbor sites. Due to the frustration, quantum Monte Carlo suffers from the sign problem and cannot provide precise results here. Recent Variational Monte Carlo [44], DMRG [45, 46], finite PEPS [47] and iPEPS [17] calculations show that the ($T = 0$) Néel phase survives up to $J_2/J_1 \simeq 0.46(2)$, where a second-order phase transition to a (possibly critical) spin liquid takes place [18, 45–47]. We choose here $J_2 = 0.5$ ($J_1$ is set to 1 as before), inside the spin liquid phase.

Since interactions occur on all plaquette diagonals, we now have to use the $a$-$b$-$c$-$d$ TS decomposition of Fig. 2(b)[4], i.e. write the infinitesimal imaginary-time propagator as

$$G(2\tau) \simeq \prod_d G_d(\tau) \prod_c G_c(\tau) \prod_b G_b(\tau) \prod_a G_a(\tau), \tag{10}$$

where the $J_1$ couplings have been equally split between adjacent plaquettes. Then, in addition to the previous two first steps associated to the $a$ and $b$ plaquettes, we now have to

---

[4]We have checked that, for $J_2 = 0$, the $a$-$b$ and $a$-$b$-$c$-$d$ decoupling schemes give identical results.

complete two more steps. After the two first steps, the optimized $\mathcal{A}''$ tensor is approximately $C_{2v}$-symmetric (instead of $C_{4v}$-symmetric, because of the extra diagonal interactions) and we (partially) correct the deviation due to the Trotter error by symmetrizing it. For the $c$ plaquette, the ($C_{2v}$-symmetrized) $\mathcal{A}''$ tensor is rotated by 90-degrees and the optimization of the fidelity leads to a new $\mathcal{A}'''$ tensor, which is rotated by 180-degrees and used for the $d$ plaquette. We observed (for $D = 7$) that the last optimized tensor after these 4 steps is almost $C_{4v}$-symmetric, as expected, and can be used (after exact symmetrization) in the single site CTMRG to compute the new environment. Note however that, for $D = 4$, the above procedure does not seem to provide reliable results. We comment in Appendix A on the role of the bond dimension.

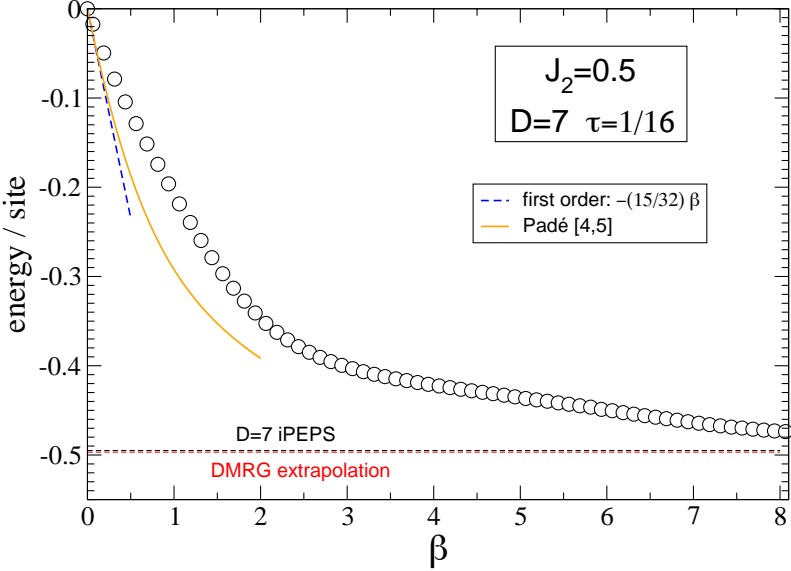

Figure 8: (a) Mean energy of the $J_1 - J_2$ model at $J_2 = 0.5$ vs $\beta = 1/T$, computed using $\mathcal{V} = 1 \oplus 0 \oplus 1$ ($D = 7$), $\chi_{\text{TE}} = D^2$ and SE. The exact 1st order high-T behavior – linear in $\beta$ (blue dotted line) – and a Padé approximation (see text) are shown for comparison. The DMRG extrapolated [45] and the $D = 7$ iPEPS [18] groundstate energies (almost indistinguishable at this scale) are also shown.

The temperature-dependance of the thermal energy versus the inverse-temperature $\beta$ is shown in Fig. 8 for $D = 7$ using a Trotter step $\tau = 1/16$. Furthermore, from the series expansion in $\beta$ up to 9th-order [48], we constructed several Padé approximants which show convergence up to $\beta \simeq 2$ (we display the specific Pade [4, 5] approximant in Fig. 8). Even at the smallest $\beta$, for which the energy is linear in $\beta$, we see deviations from the expected analytical predictions. This suggests that our iPEPO ansatz is not optimal to approximate the TDO of the frustrated model (see Appendix A for more details). We believe, either the dimension $D = 7$ of the virtual space of the iPEPO is still not large enough or the analytical iPEPO form of the TDO itself is not appropriate. In any case, we observe that, at large $\beta$, the thermal energy seems to slowly decrease towards the estimated groundstate energy [18, 45] shown on the plot, rather than to exponentially saturate to a minimum value. This may be a signature of the critical nature of the spin liquid phase [18, 45–47].

## 4 Conclusion and perspectives

Within the last ten years, tensor network methods have become the state-of-the-art technique to deal with two-dimensional correlated lattice models, and frustrated quantum spins in particular, for which QMC is inapplicable due to the sign problem. Here, we have more specifically examined the efficiency and the accuracy of tensor network techniques to compute finite-temperature properties in quantum spin systems with diverging correlation length at low temperature, a particularly challenging situation. In addition, our goal was to explicitly preserve the symmetries of the problem, namely the lattice symmetry as well as the continuous spin-rotation (SU(2)) symmetry. Thirdly, we wished to design a set up capable to deal with longer-range (frustrating) interactions. Our symmetric iPEPO ansatz of the TDO, associated to a *plaquette* TS discretized time-evolution fulfils such requirements. This framework has first been tested on the square lattice spin-1/2 Heisenberg model and confronted to large-scale QMC results (in the thermodynamic limit). First, we found that our ansatz, despite its relatively small bond dimension, is capable to generate thermal states with large correlation length, of the order of 20 lattice spacings or even more. Unfortunately, the rapid exponential growth of the Heisenberg model correlation length at low temperature is beyond the current ability of the method. Nevertheless, we showed that some observables (like the thermal energy or the staggered magnetization estimator) follow an approximate inverse-correlation length scaling, which could be of practical use for future investigations.

Our framework based on plaquette Trotterisation, allowed to also include antiferomagnetic coupling between next-nearest neighbor sites, and to investigate the long-time debated $J_1 - J_2$ quantum spin model on the square lattice. Our preliminary results show a "proof of principle" that the technique can be used in such a case. However, we found that our iPEPO does not accurately capture the correct temperature behavior. We believe a more entangled ansatz with increased bond dimension becomes necessary in the presence of magnetic frustration. This challenging issue is left for future studies.

## Acknowledgements

DP acknowledges enlightening conversations with Sylvain Capponi, Philippe Corboz and Juraj Hasik, as well as support from the TNSTRONG ANR-16-CE30-0025 and TNTOP ANR-18-CE30-0026-01 grants awarded by the French Research Council. This work was granted access to the HPC resources of CALMIP and GENCI supercomputing centers under the allocation 2017-P1231 and A0030500225, respectively. The QMC results were obtained using the looper application [39] of the ALPS project [49, 50].

## A Decomposition of the $2 \times 1$ and $2 \times 2$ gate operators

In the presence of a non-zero frustrating $J_2$ interaction, a 4-sites $2 \times 2$ gate has to be used, in contrast to the $J_2 = 0$ case where the TS decomposition could be performed in terms of $2 \times 1$ units. We believe that there is a minimum bond dimension $D$ of the PEPO in order to obtain the correct high-temperature behavior $e(\beta) = -\frac{3}{8}(J_1^2 + J_2^2)\beta$. The minimum $D$ value should be intrinsically connected to the bond dimension needed to decompose the $2 \times 1$ ($2 \times 2$) gate

operator $\mathcal{G}_{ij}^{kl}$ ($\mathcal{G}_{ijmn}^{klrs}$) in terms of the product of two (four) rank-3 (rank-4) site tensors,

$$\mathcal{G}_{ij}^{kl}(\tau) = \sum_{u=1}^{D} T_{i;u}^{k} T_{j;u}^{l} \tag{11}$$

$$\mathcal{G}_{ijmn}^{klrs}(\tau) = \sum_{u,v,w,x=1}^{D} T_{i;uv}^{k} T_{j;vw}^{l} T_{m;wx}^{r} T_{n;xu}^{s}, \tag{12}$$

where $i, j, m, n$ ($k, l, r, s$) label the physical degrees of freedom on the bottom (top) – see Fig. 3(b). Indeed, at $\beta = 0$ (infinite temperature), the TDO is just $\mathcal{L} = \mathcal{L}_0 = \mathcal{I}^{\otimes N_s}$ and maximizing the fidelity (3) amounts simply to finding for $\mathcal{L}'$ a tensor product given by the r.h.s. of (11) or (12) which best approximates (or exactly corresponds to) $\mathcal{G}$.

We first assume $J_2 = 0$ and start from a $2 \times 1$ gate. A simple singular value decomposition, gives $D = d^2 = 4$ corresponding to a $1 \oplus 0$ virtual space and

$$\mathcal{G}_{ij}^{kl} = \sum_{u=1}^{4} (T_0)_{i;u}^{k} \Delta_{uu'} (T_0)_{j;u'}^{l}, \tag{13}$$

where $T_0$ is a constant rank-3 tensor independent of the parameters $\tau$ and $J_1$ and $\Delta$ is a diagonal positive matrix,

$$\Delta = \begin{pmatrix} x & 0 & 0 & 0 \\ 0 & x & 0 & 0 \\ 0 & 0 & x & 0 \\ 0 & 0 & 0 & y \end{pmatrix}, \tag{14}$$

with $x = \frac{1}{2} e^{-\frac{J_1 \tau}{8}} \left( e^{\frac{J_1 \tau}{2}} - 1 \right)$ and $y = \frac{1}{2} e^{-\frac{J_1 \tau}{8}} \left( e^{\frac{J_1 \tau}{2}} + 3 \right)$. The site tensor $T$ in (11) can then be expressed as $T = T_0 \sqrt{\Delta}$ namely,

$$T = \begin{pmatrix} -\sqrt{x/2} & 0 & 0 & \sqrt{y/2} \\ 0 & 0 & \sqrt{x} & 0 \\ 0 & \sqrt{x} & 0 & 0 \\ \sqrt{x/2} & 0 & 0 & \sqrt{y/2} \end{pmatrix}. \tag{15}$$

When a $2 \times 2$ gate is used (as in the main text), the decomposition (12) is not exact for $D = 4$ but the optimized overlap is nevertheless very close to 1 as seen in Fig. 9(a). It is relevant to compare the fidelity to a reference obtained by choosing $\mathcal{L}' = \mathcal{L}_0$ i.e. $\mathcal{O}_{\text{ref}} = \text{Tr}(\mathcal{G}_{ijmn}^{klrs}(\tau))/\text{Tr}(\mathcal{G}_{ijmn}^{klrs}(0))$. We see that the deviation $1 - \mathcal{O}$ is almost five order of magnitude smaller than $1 - \mathcal{O}_{\text{ref}}$.

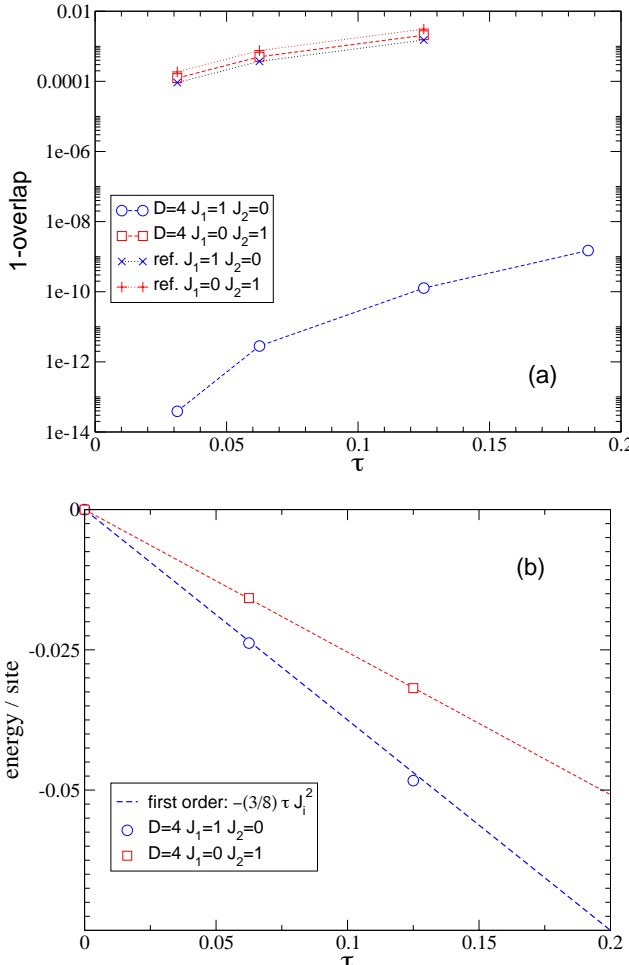

Figure 9: (a) Deviation of the maximized fidelity $\mathcal{O}$ for the first $\tau$-step at $\beta = 0$ computed on a $2 \times 2$ isolated plaquette. References (see text) are shown as $\times$ and $+$. (b) Energy per site obtained from the optimized site tensor on an isolated $2 \times 2$ plaquette.

We now turn to $J_2 \neq 0$ and, for simplicity, we choose $J_1 = 0$. As shown in Fig. 9(a), the optimized fidelity is very close to the reference one. This proves that a $D = 4$ is not large enough to capture the $\beta \to 0$ limit. To provide further evidence, we have computed the energy of the isolated $2 \times 2$ plaquette using the optimized $T$ site tensor[5]. For $J_2 = 0$, we find the exact $\tau \to 0$ asymptotic behavior $e(\tau) = -\frac{3}{8}J_1^2\tau$, as shown in Fig. 9(b). In contrast, when $J_2 \neq 0$ and $J_1 = 0$, the energy deviates substantially from the expected behavior, $e(\tau) = -\frac{3}{8}J_2^2\tau$.

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
