# Peer review of "Finite-temperature symmetric tensor network for spin-1/2 Heisenberg antiferromagnets on the square lattice"

_SciPost Physics, doi:SciPost Phys. 10, 019 (2021)_

## Round 1 · Referee Report · Anonymous (Referee 1) · 2020-11-8

Report

A construction of a double layer tensor network is proposed for finite temperature 2-dimensional quantum spin models. The SU(2) symmetry of the systems are naturally represented by the choice of local symmetric tensors. For the update of local tensors with respect to the imaginary time evolution, the corner transfer matrix renormalization group (CTMRG) method is used. In addition to the faithful treatment of environment, a simplified scheme is also proposed. Efficiency of the proposed construction of the density matrix is checked by its application to S=1/2 Heisenberg antiferromagnets. Qualitative agreement with the quantum Monte-Carlo simulation is observed. As a numerical approximation scheme, I recommend the publication of this manuscript, if the following points are improved.

(1) Concerning to the spontaneous magnetization of the 2D Heisenberg model, there are recent precise estimations in addition to the references 1-3. Anderson's contribution is also important.

(2) Baxter's articles can be cited as references on CTMRG. Reference 31 introduced a way of constructing the environment tensor by use of CTMRG.

(3) Physical background of introducing the double layer tensor network, instead of single layered one, should be explained. Probably the reason is that the ancillas carry only weak entanglement in very low temperature.

(4) The definition of ¥cal L should be given in the main text. One has to look at figures and captions in order to understand the definition.

(5) The environment tensor is constructed by means of the original uniform tensor network. This is an approximation scheme, since after the operation of ¥cal L, the period of translational invariance becomes two lattice constants. Please look at the treatment in Ref. 22.

(6) If the efficiency of the simplified environment is checked for the case D=4, one can somehow capture the reliability of the data obtained for D=7 under the simplified numerical treatment.
  • validity: good
  • significance: good
  • originality: high
  • clarity: good
  • formatting: reasonable
  • grammar: excellent

Author:  Didier Poilblanc  on 2020-12-10  [id 1069]

(in reply to Report 1 on 2020-11-08)

We thank the Referee for his/her careful reading. Based on his/her relevant comments we have (slightly) modified the manuscript accordingly:

(1) We have added new references to Anderson and to Calandra & Sorella.

(2) We have added new references to Baxter and to Nishino et al.

(3) We believe one important (mathematical) reason is the positivity of the ansatz which is guarantied "for free" (as stated in the manuscript). The positivity is a very important physical feature of the TDO. At very low-temperature, however, the ansatz is not particularly good (as shown e.g. by deviation from QMC for J2=0), and the usual iPEPS gives better \beta->0 energy for the same bond dimension.

(4) Good point. The symbol (\cal L) of the iPEPO is now provided in sec. 2.1 together with a proper definition in the text.

(5) We agree that, strictly speaking, the environment is not fully transitionally invariant before the 2nd step (it IS however before the 1st-step) . We believe however that the error introduced by assuming a transitionally invariant environment in the 2nd step are very small, in particular because a 2x2 plaquette ST decomposition - 2 step-only - framework is used and also because a C4v-symmetrization is performed after the second step, partially correcting various sources of error (like the TS finite \tau error, etc...).

(6) Good point indeed. This is why both the SE and the FE data for D=4 have been compared in Fig. 5(b). For \beta=1, the error of the SE remains quite small, but is not negligible. That should also be true for D=7.

---

## Round 1 · Referee Report · Jürgen Schnack (Referee 2) · 2020-11-26

Strengths

This is a valuable paper on a further improvement of the tensor network approach to thermal observables of two-dimensional quantum spin lattices. In particular, I appreciate that the newly developed iPEPO method is compared to the square lattice that can be treated exactly by means of QMC. A comparison to the frustrated J1-J2 square lattice shows that the current version is not yet capable of reliably treating such frustrated problems.

Weaknesses

The paper is rather technical, even though formulas are avoided. But this is probably intrinsic to the topic.

Report

This is a very good state of the art paper on a pressing topic. I recommend publication after correction of a few typos.

Requested changes

  1. Physicists sloppily use "finite" although "non-zero" would be correct. I think this applies to all "finite" in this paper. Please correct.

  2. 2nd line abstract - Operators; 2nd para on page 1 - dimensionnal - delete one n; 3rd para on page 1 - has been.

  3. Explain notation $0\oplus 1[4]$ used here and elsewhere.

  4. Page 3, 5th line from below, "... contacted with each of them." Not clear. Who with whom?

  5. Figure 6 too big, could not see caption completely.

  6. Figure 6, an arrow showing how $\beta$ grows would be nice in (a) and (b). This helps to understand the QMC data in (a) for the various $\beta$.

  7. Second para of 3.2, formula runs over right page boundary.

  8. Please take care of capital letters in the references such as Heisenberg etc.

  • validity: high
  • significance: high
  • originality: high
  • clarity: good
  • formatting: reasonable
  • grammar: excellent

Author:  Didier Poilblanc  on 2020-12-10  [id 1070]

(in reply to Report 2 by Jürgen Schnack on 2020-11-26)

Dear Prof. Schnack,

We appreciate your careful reading of the manuscript. We have provided the following corrections according to your remarks:

(1) We have changed "finite temperature" into "non-zero temperature" but we have kept "finite size effects" since, in the later case, it means "not infinite" !

(2) Typos corrected - thanks !

(3) We have added "...and the physical Hilbert space becomes the direct sum of SU(2) multiplets such as ..." at the appropriate place in the text. A clarification has also been added in the caption of figure 1.

(4) We have rephrased as: "...two independent subclasses of $A$ tensors obtained by contracting each of the two rank-3 tensors with the appropriate (matching) rank-5 tensors. "

(5) All figures have been reduced in size. The caption of figure 6 should be visible now.

(6) We have added "$\beta=1,1.5$ and $2$ (QMC) data displayed from bottom to top" in the caption of figure 6.

(7) This formula has been displayed as a numbered equation now.

(8) Done. Thanks.

Best wishes,
Didier Poilblanc

---

## Round 1 · Referee Report · Anonymous (Referee 3) · 2020-12-2

Report

In this paper the authors perform finite temperature simulations of the square lattice Heisenberg model and the frustrated J1-J2 Heisenberg model using infinite projected entangled pair states (iPEPS). They introduce a plaquette based imaginary time evolution approach based on an iPEPS ansatz which preserves both the spin SU(2) symmetry and the C_{4v} symmetry of the lattice. Two variants are considered: a full environment (FE) approach where the environment is fully taken into account (controlled by the bond dimension \chi), and a "simple" environment approach (SE) which is computationally cheaper, but where the environment is only partially taken into account. For the Heisenberg model the authors find good agreement with benchmark QMC data down to a certain temperature, beyond which the data starts to deviate due to finite bond dimension effects and limitations of the SE scheme. The iPEPS data for the J1-J2 model deviate from high-temperature series expansion results already at high temperature, based on which the authors conclude that larger bond dimensions are required in this case in order to accurately capture the finite temperature behavior.

In my opinion, this work provides a significant and interesting contribution to the development of 2D tensor approaches at finite temperature. The benchmarks with QMC are valuable to illustrate the strengths and weaknesses of the approach. Interesting new features of this work, which have not been explored before at finite temperature, are the implementation of SU(2) spin and C_{4v} lattice symmetric tensors, and the plaquette-type update FE/SE approaches. The preliminary results for the J1-J2 may be somewhat disappointing, but there is still room for improvement, such that also this more challenging case may be within reach in future. Besides this, the paper is written in a clear way.

For these reasons I can recommend this paper to be published in Scipost Physics, after the authors have addressed the points listed below.

Comments and questions ———————————-

1) The deviation between iPEPS and high-T series expansion result even at high temperature is rather surprising at first sight. But in principle it is indeed possible that, because of the additional interactions, there is a sizable finite D error already after one time evolution step. Have the authors compared the D=4 and D=7 results at high temperature, i.e. are there strong finite D effects already visible at high temperature? That would be a useful crosscheck to see whether the finite D is indeed the problem here.

2) One comment regarding the sentence at the bottom of page 1: "These methods are based on a smart parametrization of the thermal density operator (TDO) in terms of a double-layer Projected Entangled Pair Operator (PEPO), which naturally guarantees its positivity." This is true in most of the previous works, but not for Refs. [17, 20] where only a single layer was used. I would be good to rephrase this accordingly.

3) On page 5, when introducing CTMRG, the initial reference of the CTMRG method should also be added: T. Nishino and K. Okunishi, J. Phys. Soc. Jpn. 65, 891 (1996).

4) Do I understand it correctly that in both the FE and SE approach the same CTMRG method is used (with a full chi) and only in the last CTMRG step the chi is truncated down to 1 in the SE approach in order to compute the overlaps in a cheaper way? One comment: I guess there is a lot of room for approaches in between FE and SE, e.g. starting the CG with the cheaper SE and gradually increase the chi towards the FE to speed up the convergence while still being computationally cheaper than the FE-only approach.

5) There is a problem with the caption of Fig.6: it is cut and not all the text is displayed. Also one inline formula at the bottom of page 11 is not well formatted and not fully displayed.

6) It may be good to change the layout such that the Figures 7, 8 appear closer to the location in the text where they are mentioned (overall I think the figure sizes could be reduced).

7) Minor typos: page 1, second paragraph, second line: "two-dimensionnal" -> " two-dimensional" page 4, second paragraph, second line: a space is missing in " the"infinitesimal" "

  • validity: -
  • significance: -
  • originality: -
  • clarity: -
  • formatting: -
  • grammar: -

Author:  Didier Poilblanc  on 2020-12-28  [id 1114]

(in reply to Report 3 on 2020-12-02)

We thank the Referee for insightful comments. We list below our replies to each point, one by one:

(1) Indeed, for J2 non-zero, the method at D=7 was unable to reproduce the linear high-temperature behavior well. In fact, calculations with D=4 have also been attempted (but not reported in the initial version of the paper) and unphysical results have been obtained. We believe there is a minimum D to capture the high-temperature limit exactly. For J2=0, it is easy to see that the 2-site (time-evolution) gate can be exactly split (by SVD) introducing D=d^2=4 (V=1+0) virtual degrees of freedom (see new appendix added). For the 4-site (square) gate at J2 non-zero, we believe that a higher-order factorisation into four tensors contracted on the square edges should involve a significantly larger virtual space of dimension D, even when expending the time-evolution operator in first order. This is discussed in the new appendix.

(2) We have rephrased this part following the Referee comment : “These methods are based on a parametrization of the thermal density operator (TDO) in terms of a Projected Entangled Pair Operator (PEPO). Among the recent developments, the most efficient framework uses a double-layer PEPO, instead of a single layer one,
which naturally guarantees the positivity of the approximated TDO.”

(3) Indeed, a reference to Nishino et al. is appropriate. In addition, a reference to Baxter has also been added.

(4) It is indeed true that one can interpolate between the SE and the FE cases, by just gradually increasing the number \chi_overlap of highest weights of the fixed-point corner matrix used in the overlap cost-function, from 1 to \chi_TE. We have added a comment in subsection 2.3.3. Note that we have not attempted to consider such an intermediate scheme yet, it is left for future studies on the frustrate model (see (1)).

(5) We fixed the problem with Fig. 6 (reducing the sizes of all figures a bit) and the formula on page 11 (now displayed as a numbered equation).

(6) All figure sizes have been reduced as suggested and, we believe, the layout is improved.

(7) Thank you for mentioning the typos.

Author:  Didier Poilblanc  on 2020-12-22  [id 1102]

(in reply to Report 3 on 2020-12-02)

We thank the Referee for insightful comments. We list below our replies to each point, one by one:

(1) Indeed, for J2 non-zero, the method at D=7 was unable to reproduce the linear high-temperature behavior well. In fact, calculations with D=4 have also been attempted (but not reported in the initial version of the paper) and unphysical results have been obtained. We believe there is a minimum D to capture the high-temperature limit exactly. For J2=0, it is easy to see that the 2-site (time-evolution) gate can be exactly split (by SVD) introducing D=d^2=4 (V=1+0) virtual degrees of freedom (see new appendix added). For the 4-site (square) gate at J2 non-zero, we believe that a higher-order factorisation into four tensors contracted on the square edges should involve a significantly larger virtual space of dimension D, even when expending the time-evolution operator in first order. This is discussed in the new appendix.

(2) We have rephrased this part following the Referee comment : “These methods are based on a parametrization of the thermal density operator (TDO) in terms of a Projected Entangled Pair Operator (PEPO). Among the recent developments, the most efficient framework uses a double-layer PEPO, instead of a single layer one,
which naturally guarantees the positivity of the approximated TDO.”

(3) Indeed, a reference to Nishino et al. is appropriate. In addition, a reference to Baxter has also been added.

(4) It is indeed true that one can interpolate between the SE and the FE cases, by just gradually increasing the number \chi_overlap of highest weights of the fixed-point corner matrix used in the overlap cost-function, from 1 to \chi_TE. We have added a comment in subsection 2.3.3. Note that we have not attempted to consider such an intermediate scheme yet, it is left for future studies on the frustrate model (see (1)).

(5) We fixed the problem with Fig. 6 (reducing the sizes of all figures a bit) and the formula on page 11 (now displayed as a numbered equation).

(6) All figure sizes have been reduced as suggested and, we believe, the layout is improved.

(7) Thank you for mentioning the typos.

---

## Round 2 · Referee Report · Anonymous (Referee 1) · 2021-1-5

Strengths

Symmetry in the target physical system is considered from the structure of local tensors. This construction greatly reduce the number of parameters considered in numerical optimization.

Weaknesses

Since the approach fits to systems with high symmetry, generalization to systems with lower symmetry or random systems is not straight forward.

Report

The authors have properly revised the manuscript. Now the explanation of Figures 3 and 4 are satisfactory. Thus I recommend the publication of this article.

---

## Round 2 · Referee Report · Jürgen Schnack (Referee 2) · 2021-1-6

Report

Accept the revised manuscript

---

## Round 2 · Referee Report · Anonymous (Referee 3) · 2021-1-10

Report

The authors have addressed all the points in the revised version and I can thus recommend publication of this paper.

---

## Round 2 · List of Changes

All changes have been described in the detailed answers to the 3 referee reports.

---

## Editorial Decision

published